# Reduction of elevated proton leak rejuvenates mitochondria in the aged cardiomyocyte

Huiliang Zhang[1], Nathan N Alder[2], Wang Wang[1,3], Hazel Szeto[4], David J Marcinek[5], Peter S Rabinovitch[1]*

[1]Department of Laboratory Medicine and Pathology, University of Washington, Seattle, United States; [2]Department of Molecular and Cell Biology, University of Connecticut, Storrs, United States; [3]Mitochondria and Metabolism Center, Department of Anesthesiology and Pain Medicine, University of Washington, Seattle, United States; [4]Social Profit Network Research Lab, Alexandria LaunchLabs, New York, United States; [5]Department of Radiology, University of Washington, Seattle, United States

**Abstract** Aging-associated diseases, including cardiac dysfunction, are increasingly common in the population. However, the mechanisms of physiologic aging in general, and cardiac aging in particular, remain poorly understood. Age-related heart impairment is lacking a clinically effective treatment. Using the model of naturally aging mice and rats, we show direct evidence of increased proton leak in the aged heart mitochondria. Moreover, our data suggested ANT1 as the most likely site of mediating increased mitochondrial proton permeability in old cardiomyocytes. Most importantly, the tetra-peptide SS-31 prevents age-related excess proton entry, decreases the mitochondrial flash activity and mitochondrial permeability transition pore opening, rejuvenates mitochondrial function by direct association with ANT1 and the mitochondrial ATP synthasome, and leads to substantial reversal of diastolic dysfunction. Our results uncover the excessive proton leak as a novel mechanism of age-related cardiac dysfunction and elucidate how SS-31 can reverse this clinically important complication of cardiac aging.

*For correspondence:
petersr@uw.edu

## Introduction

Mitochondria are both the primary source of organismal energy and the major source of cellular reactive oxygen species (ROS) and oxidative stress during aging (*Dai et al., 2014*). Aged cardiac mitochondria are functionally changed in redox balance and are deficient in ATP production (*Lesnefsky et al., 2016*). Numerous reported studies have focused on redox stress and ROS production in aging (*Dai et al., 2014*). However, in its simplistic form, the free radical theory of aging has become severely challenged (*Pérez et al., 2009*).

While more attention has been placed on mitochondrial electron leak and consequent free radical generation, proton leak is a highly significant aspect of mitochondrial energetics, as it accounts for more than 20% of oxygen consumption in the liver (*Brand, 2005*) and 35–50% of that in muscle in the resting state (*Rolfe and Brand, 1996*). There are two types of proton leak in the mitochondria: (1) constitutive, basal proton leak, and (2) inducible, regulated proton leak, including that mediated by uncoupling proteins (UCPs) (*Divakaruni and Brand, 2011*). In skeletal muscle, a majority of basal proton conductance has been attributed to adenine nucleotide translocase (ANT) (*Brand et al., 2005*). Although aging-related increased mitochondrial proton leak was detected in the mouse heart, kidneys, and liver by indirect measurement of oxygen consumption in isolated mitochondria (*Harper et al., 1998*; *Serviddio et al., 2007*), direct evidence of functional impact remains to be

further investigated. Moreover, the exact site and underlying mechanisms responsible for aging-related mitochondrial proton leak are unclear.

SS-31 (elamipretide), a tetrapeptide (D-Arg-2′,6′-dimethyltyrosine-Lys-Phe-NH2), binds to cardiolipin-containing membranes (*Birk et al., 2013*) and improves cristae curvature (*Szeto, 2014*). Prevention of cytochrome *c* peroxidase activity and release has been proposed as its major basis of activity (*Szeto, 2014*; *Szeto and Birk, 2014*). SS-31 is highly effective in increasing resistance to a broad range of diseases, including heart ischemia reperfusion injury (*Cho et al., 2007*; *Szeto, 2008*), heart failure (*Dai et al., 2013*), neurodegenerative disease (*Yang et al., 2009*), and metabolic syndrome (*Anderson et al., 2009*). In aged mice, SS-31 ameliorates kidney glomerulopathy (*Sweetwyne et al., 2017*) and brain oxidative stress (*Hao et al., 2017*) and has shown beneficial effects on skeletal muscle performance (*Siegel et al., 2013*). We have recently shown that administration of SS-31 to 24-month-old mice for 8 weeks reverses the age-related decline in diastolic function, increasing the E/A from just above 1.0 to 1.22, restoring this parameter 35% toward that of young (5-month-old) mice (*Chiao et al., 2020*). However, how SS-31 benefits and protects aged cardiac cells remains unclear.

In this report, we investigated the effect and underlying mechanism of action of SS-31 on aged cardiomyocytes, especially on the mitochondrial proton leak. Using the naturally aged rodent model we provided direct evidence of increased proton leak as the primary energetic change in aged mitochondria. We further show that the inner membrane protein ANT1 mediates the augmented proton entry in the old mitochondria. Most significantly, we demonstrate that SS-31 acutely prevents the excessive mitochondrial proton entry and rejuvenates mitochondrial function through direct association with ANT1 and stabilization of the ATP synthasome.

## Results

### SS-31 alleviates the excessive mitochondrial proton leak in old cardiomyocytes

To examine whether SS-31 restores aging mitochondrial function, we applied the Seahorse mitochondrial stress assay to intact primary cardiomyocytes in a non-working state (not paced for contraction). The Seahorse assay revealed higher mitochondrial basal respiration in cells from old mice than that in young mouse cells (*Figure 1A,C*); however, the maximal respiratory rate was not significantly different (*Figure 1A,D*). The increased basal respiration was attributable to a higher proton leak in old cardiomyocytes ($164 \pm 16$ in 24 month vs $82 \pm 12$ in young, pmol/min/800 cells, n = 7–14, p<0.01) (*Figure 1A,B*). Although, SS-31 has only a minor and non-significant effect on young cardiomyocytes (*Figure 1—figure supplement 1*), acute in vitro treatment of isolated old cardiomyocytes with SS-31 (100 nM, 1 µM, or 10 µM for 2 hr), caused reduced mitochondrial proton leak (*Figure 1A, B* and *Figure 1—figure supplement 2*), shifting their respiratory pattern closer to that of young cells. These results indicate that SS-31 directly protects aging cardiac energetics through rapid rejuvenation of mitochondrial respiration in cardiomyocytes, and in particular, by reducing proton leak.

### SS-31 restores the resistance to external pH gradient stress in mitochondria of old cardiomyocytes

The evaluation of mitochondrial proton leak by Seahorse assay is indirect, as it is based on the oxygen consumption rate. Thus, to directly investigate the reduction of mitochondrial proton leak in old cardiomyocytes by SS-31, we expressed the protein mt-cpYFP, a mitochondrial matrix-targeted pH indicator (*Demaurex and Schwarzländer, 2016*; *Schwarzländer et al., 2012*; *Wang et al., 2016a*; *Wei-LaPierre et al., 2013*) in the rat cardiomyocytes (*Figure 2—figure supplement 1*). Taking advantage of the pH sensitive character of mt-cpYFP, we developed a novel protocol to evaluate mitochondrial proton leak by exposing mitochondria to a pH gradient stress in saponin permeabilized, mt-cpYFP expressing cardiomyocytes (*Figure 2A,B*). The drop in mt-cpYFP 488/405 ratio is due to proton leak through the mitochondrial inner membrane into the mitochondrial matrix. To evaluate the physical properties of the mitochondrial inner membrane in the absence of mitochondrial activity, we permeabilized the cardiomyocytes in a buffer that contained no substrates, ATP, or ADP. We found that aging reduced cardiomyocyte mitochondrial resistance to a proton gradient stress (*Figure 2A,B*). More importantly, we found 10 µM SS-31 treatment in vitro restored

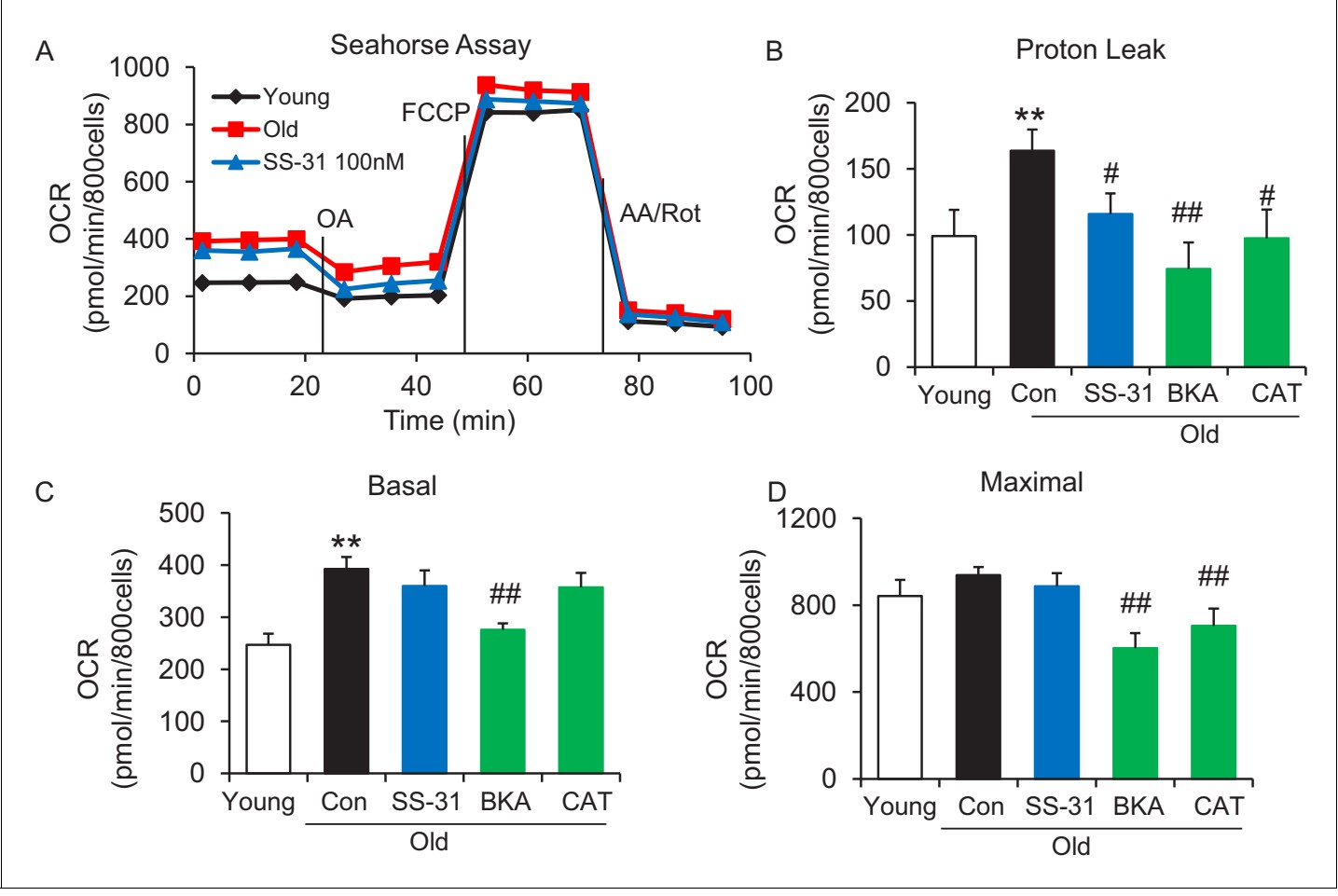

**Figure 1.** SS-31 alleviates the excessive mitochondrial proton leak of cardiomyocytes from 24 mo old mice. (A) Representative Seahorse assay traces of cardiomyocytes isolated from untreated young and old mice, then exposed or not to 100 nM SS-31 for 2 hr in vitro. Aging-increased basal respiration (C), which was attributable to the augmentation of proton leak (B), but did not affect maximal respiration (D). ANT1 inhibitors Bongkrekic acid (BKA, 10 µM) and carboxyatractyloside (CAT, 20 µM) 2 hr treatment decreased the proton leak (B) and basal respiration (C) but also decreased maximal respiration (D) in old cardiomyocytes. N = 5–14 mice in each group; one-way ANOVA followed by Fisher's LSD test. *p<0.05, **p<0.01 vs young; #p<0.05, ##p<0.01 vs old controls.

The online version of this article includes the following figure supplement(s) for figure 1:

**Figure supplement 1.** SS-31 has a minor effect on mitochondrial respiration in the young cardiomyocytes.

**Figure supplement 2.** SS-31 reaches its inhibitory effect on proton leak suppression at low concentrations.

cardiomyocyte mitochondrial inner membrane resistance to the pH gradient stress in the aged cardiomyocytes (*Figure 2A,B*). SS-31 treatment largely prevented the decline in matrix pH of old cells after the external pH was reduced to 6.9 (*Figure 2—figure supplement 2*) and 5.3 (*Figure 2B,C*) and slowed the rate of cpYFP 488/405 change after pH 6.9 (*Figure 2—figure supplement 3*, *Figure 2D*). At pH 4.5 SS-31 continued to enhance resistance to proton permeability in the treated old cells. This is unlikely to represent a biological benefit at this nonphysiologic pH, but does indicate the substantial change in physical properties of the inner membrane after interaction with SS-31 (*Figure 2B*). To further evaluate the kinetics of SS-31 effect on mitochondrial proton permeability, we analyzed cpYFP fluorescence ratios at various times after exposure of the saponin treated cardiomyocytes to 10 µM SS-31. SS-31 protection on the mitochondrial matrix proton entry became significant and near maximal after 7–10 min of SS-31 treatment (*Figure 2E*). We examined the dose effect of SS-31 on proton permeability and found near-maximal effects at 100 nM SS-31 (*Figure 2F*). In summary, this is the first direct evidence that aging increases mitochondrial inner membrane proton permeability in aged cardiomyocytes and that SS-31 protects mitochondria from this proton leak.

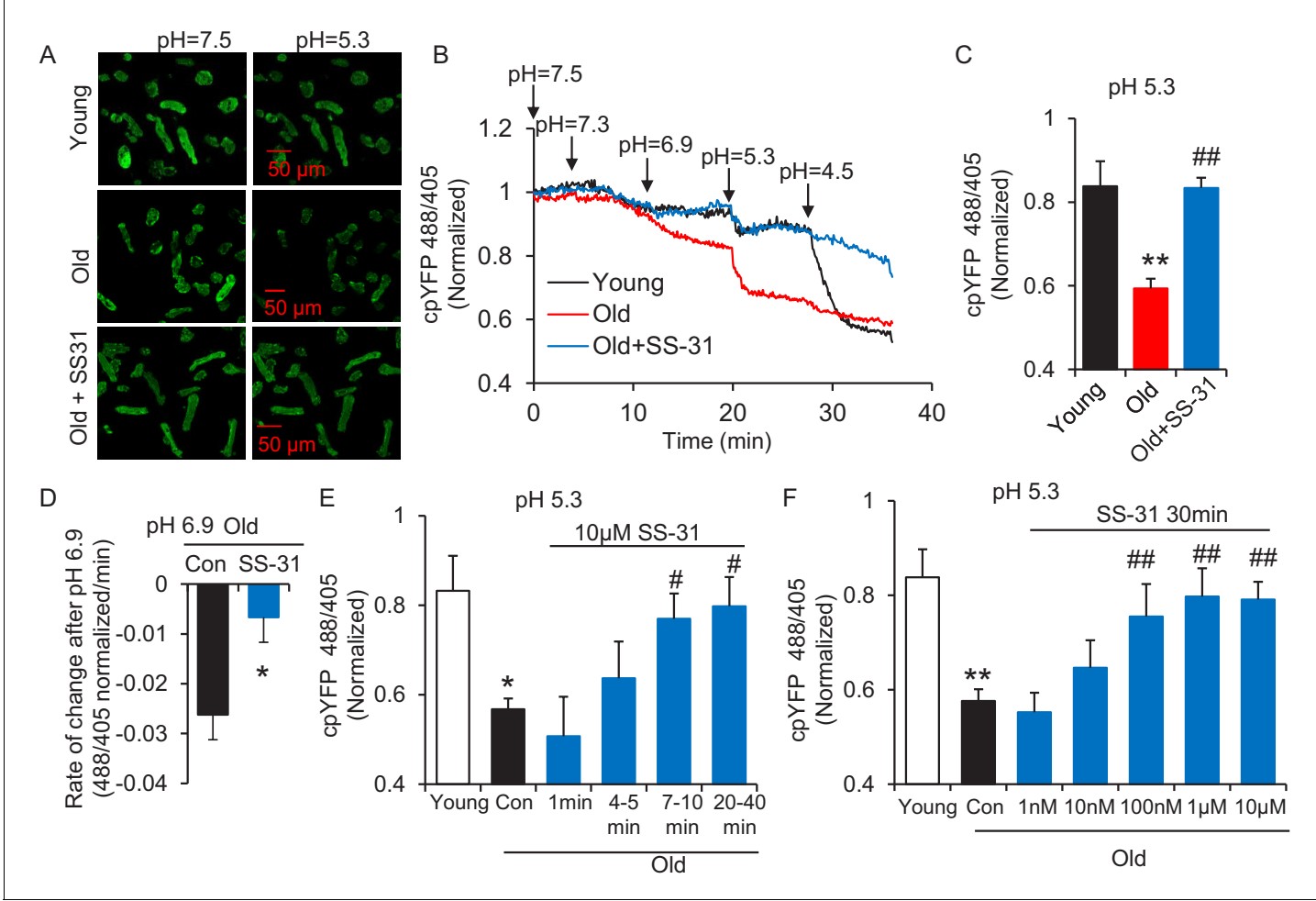

**Figure 2.** SS-31 restores the resistance of cardiomyocytes from old rat to proton entry into the mitochondrial matrix during external pH gradient stress. (A) Typical image of the effects of pH gradient stress on permeabilized rat cardiomyocyte mt-cpYFP fluorescence. Upper panel, young, middle panel, old, lower panel, old+SS-31 (10 µM, 3 days) visualized after exposure of the cells to pH 7.5 and, later, to pH 5.3. The excitation is 488 nm and collection is at 505–730 nm. (B) Saponin (50 µg/ml) permeabilized cardiomyocytes expressing mt-cpYFP were exposed to progressively lower external pH. Proton permeability of old mitochondria was greater than that of young mitochondria, but preincubation of old cells with 10 µM SS-31 for 3 days enhanced the mitochondrial inner membrane resistance to the pH stress. The traces were averaged from 4 to 19 experiments. The arrows indicate the changes of pH. (C) Quantitation of the SS-31 treatment effect on the mitochondrial matrix cpYFP ratio at pH 5.3. The data are from 7 to 8 min after the pH was adjusted to 5.3. The mean value of the normalized 488/405 ratio in old group is 0.59 with a SD ±0.10 and the range is from 0.46 to 0.77. N = 5–19 rats in each group, analyzed by one-way ANOVA with Dunnett's test. (D) SS-31 decreased the rate of cpYFP 488/405 ratio drop at pH 6.9. N = 4–10 rats in each group. The rate is calculated as indicated in *Figure 2—figure supplement 3*. Student's t-test was applied to determine the statistical significance. The time dependence (N = 3–4 rats in each group) (E) and dose dependence (N = 3–14 rats in each group) (F) of SS-31 protection of mitochondrial resistance to pH gradient stress are shown and analyzed by one-way ANOVA with Fisher's LSD test. After cardiomyocyte permeabilization, 10 µM SS-31 was added for the times shown in (E) or at the doses shown in (F) for 30 min, followed by pH stress. *p<0.05, **p<0.01 vs Young; #p<0.05, ##p<0.01 vs Old.

The online version of this article includes the following figure supplement(s) for figure 2:

**Figure supplement 1.** pH calibration of mt-cpYFP in adult cardiac myocytes with Nigericin.

**Figure supplement 2.** SS31 restores the resistance to external pH gradient stress in the old cardiomyocytes at pH 6.9.

**Figure supplement 3.** Method of quantitation of the slope of mt-cpYFP fluorescence ratio change after permeabilized cells are exposed to external pH 6.9.

# ANT1 inhibitors restore resistance to proton leak of mitochondria in old cardiomyocytes

In search of the source of the uncoupled mitochondrial proton leak in the aged cells, we examined possible involvement of proton leakage through ATPase and mitochondrial UCPs. As expected, the ATPase inhibitor Oligomycin A failed to inhibit the proton leak in pH challenged permeabilized aged cells (*Figure 3A,B*). Levels of UCP2, which is the dominant isoform of UCPs in the heart, do not change with age in hearts (*Figure 3—figure supplement 1*). Genipin, an inhibitor of UCP2, showed no effect on the proton leak in permeabilized aged cells (*Figure 3A,B*). These results suggest that ATPase and UCP2 may not be the source of the excess proton leak in the aged hearts.

Recently, the inner membrane protein ANT1 (also called AAC) was identified as the major site of proton leak in mitochondria of multiple tissues (*Bertholet et al., 2019*), and was shown to contribute to the majority of the proton leak in muscle cells (*Brand et al., 2005*). Treatment of old cardiomyocytes with either the ANT1 inhibitor bongkrekic acid (BKA) (*Ruprecht et al., 2019*) or carboxyatractyloside (CAT) (*Pebay-Peyroula et al., 2003*) completely suppressed the excess proton leak in the Seahorse assay, though unlike SS-31, they also decreased the maximal respiratory rate (*Figure 1B, D*), which is consistent with the effect seen in the ANT triple knockout model (*Karch et al., 2019*). We treated permeabilized old cardiomyocytes with the ANT1 inhibitors and examined the mt-cpYFP response to an external pH gradient using the protocol described above. BKA suppressed the proton leak in old cardiomyocytes, evidenced by the preserved 488/405 ratio at pH 5.3 and a slower 488/405 ratio decrease at pH 6.9 (*Figure 3A,B*). Similar inhibition was found with CAT treatment (*Figure 3A,B*). Taken together, these data implicate ANT1 as the major site of proton leak in aging hearts.

Considering that the inorganic phosphate carrier might be a potential source of the elevated proton entry in old mitochondria, we examined the proton permeability of old mitochondria to pH stress in a buffer that did not contain phosphate. In this buffer, the proton entry characteristics of mitochondria from old cells was indistinguishable from that of mitochondria stressed in the presence of inorganic phosphate (*Figure 3—figure supplement 2*).

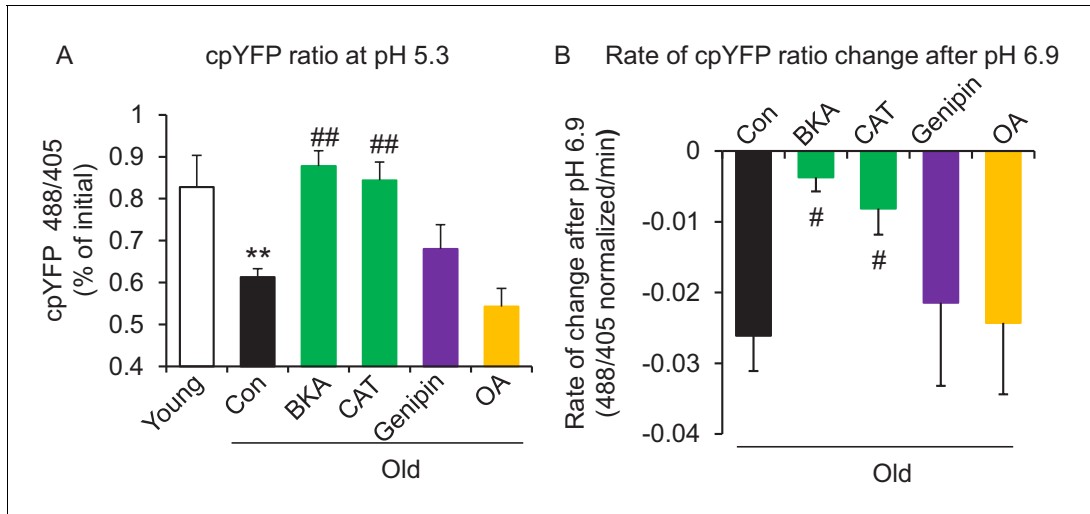

**Figure 3.** ANT1 inhibitors restore resistance to proton leak of mitochondria in old rat cardiomyocytes. ANT1 inhibitors 10 µM Bongkrekic Acid (BKA) and 20 µM carboxyatractyloside (CAT), but not 50 µM Genipin (UCP2 inhibitor) or 1 µM oligomycin A (OA, ATPase inhibitor), protected the mitochondrial matrix from decreased pH after exposure to external pH 5.3 (N = 4–19 rats in each group) (A) and reduced the rate of 488/405 decline after exposure to pH 6.9 (N = 4–10 rats in each group) (B). BKA, CAT, Genipin, or OA were added immediately after the mitochondria permeabilization. One-way ANOVA with Fisher's LSD test was applied. *p<0.05, **p<0.01 vs Young, #p<0.05, ##p<0.01 vs Old.

The online version of this article includes the following figure supplement(s) for figure 3:

**Figure supplement 1.** Aging effect on ANT1 and UCP2 cardiac protein abundance.
**Figure supplement 2.** Mitochondrial proton entry in buffer without phosphate.

## SS-31 attenuates the excessive mitochondrial flash (mitoflash) activity of aged cardiomyocytes, while normalizing membrane potential and ROS

The mitoflash (*Feng et al., 2017*; *Hou et al., 2014*; *Shen et al., 2014*; *Wang et al., 2008*; *Wang et al., 2016b*; *Zhang et al., 2015*) is triggered by nanodomain proton influx into the mitochondrial matrix (*Wang et al., 2016c*). Thus, we wondered whether the increased proton leak in the old cells triggered excessive mitoflash activity. We evaluated mitoflash activity in isolated young and old rat cardiomyocytes using the indicator mt-cpYFP, as established in the previous studies noted above. The mitochondrial mitoflash activity in the cells from old (26 mo) cardiomyocytes was higher than that of young (five mo) cells (2.8 ± 0.3 in old vs 1.4 ± 0.2,/1000 μm$^2$/100 s in young cells, n = 28–88, p<0.05). Confirming this, we detected an increase in mitoflash activity in Langendorff perfused intact aged hearts from mt-cpYFP transgenic mice (*Figure 4—figure supplement 1*). 1 hr treatment with SS-31 normalized the mitoflash activity in old cells to the young cell level (*Figure 4D*). The mitochondrial ANT1 inhibitors BKA and CAT showed super-suppression of the flash activity, reducing this frequency to half of that of young cells (*Figure 4D*). These data support the notion that proton leak from ANT1 triggers the mitoflash in cardiomyocytes and is responsible for the excess mitoflash activity of old cells. Moreover, the mitochondrial membrane potential, which is generally lower in old cardiomyocytes (*Serviddio et al., 2007*), is restored to youthful levels by SS-31 treatment (*Figure 4E*). Also, SS-31 reduced ROS production in the aged cardiomyocytes (*Figure 4F*). Thus, the reduction of mitochondrial proton leak by SS-31 is accompanied by a more youthful membrane potential and dynamic function (mitoflash), as well as less oxidative stress.

## SS-31 reverses increased mPTP opening in aged cardiomyocytes

Due to the close link previously established between the mitoflash and mitochondrial permeability transition pore (mPTP) opening (*Hou et al., 2014*), we evaluated mPTP activity by the photon-triggered mPTP opening protocol (*Figure 5A*; *Zorov et al., 2000*). Consistent with previous reports in isolated mitochondria (*Hafner et al., 2010*), we found that the time to mPTP opening is decreased in intact old cardiomyocytes (*Figure 5B*). SS-31 and the ANT1 inhibitor BKA, which stabilizes the ANT1 in the m-state open toward the mitochondrial matrix, both protect the aging-increased mitochondrial mPTP opening rate (*Figure 5B*), consistent with previous observations that BKA prevents the onset of the permeability transition (*Halestrap et al., 1997*). The ANT1 inhibitor CAT, which stabilizes ANT1 in the c-state open toward the cytosol, failed to prevent the rapid opening of the mPTP in old cells (*Figure 5B*), consistent with previous observations that it facilitates mPTP opening (*Halestrap et al., 1997*). These data indicate that SS-31 decreases mPTP opening in old cardiomyocytes.

## SS-31 associates directly with ANT1 and the ATP synthasome

To further investigate the mechanism of SS-31 protection on the proton leak, we used biotinylated SS-31 to evaluate whether SS-31 directly interacts with the ANT1 protein. Hearts were disrupted by douncing, after a low speed spin to remove fragments, mitochondria were collected by high speed spin and disrupted in digitonin, to create lipid rafts containing their associated proteins, a protocol commonly used to prepare mitochondrial supercomplexes (*Johnson et al., 2013*). This preparation was incubated with SS-31-biotin or biotin only, followed by incubation with streptavidin beads. After washing, the bead-bound fraction was eluted with excess SS-31 and analyzed by Western blotting. Biotin-SS-31 pulled down ANT1, and free SS-31 competed with the biotin-SS-31 binding to ANT1 (*Figure 6A,B*, *Figure 6—figure supplement 1*). Most notably, both BKA and CAT inhibited binding of biotin-SS-31 to ANT1 (*Figure 6A,B*). This competition was observed even at BKA and CAT concentrations in the tens of nanomolar range (data not shown), which is consistent with their reported Kd of binding to ANT1 (*Vignais et al., 1976*). Biotin-SS-31 pulldown of ANT1 was not inhibited by Genipin or Oligomycin A (*Figure 6A,B*). These data indicate that SS-31 associates closely with the ANT1 protein. Moreover, native gel and ATPase blot analysis showed that SS-31 stabilized the ATP synthasome, of which ANT1 and ATPase are critical members (*Ko et al., 2003*; *Figure 6D,E*). However, SS-31 treatment did not produce a detectable increase in mitochondrial complex proteins by Coomassie blue staining (*Figure 6C*). Taken together, these data suggest that SS-31 interacts directly with ANT1 and stabilizes the ATP synthasome in old cardiomyocytes.

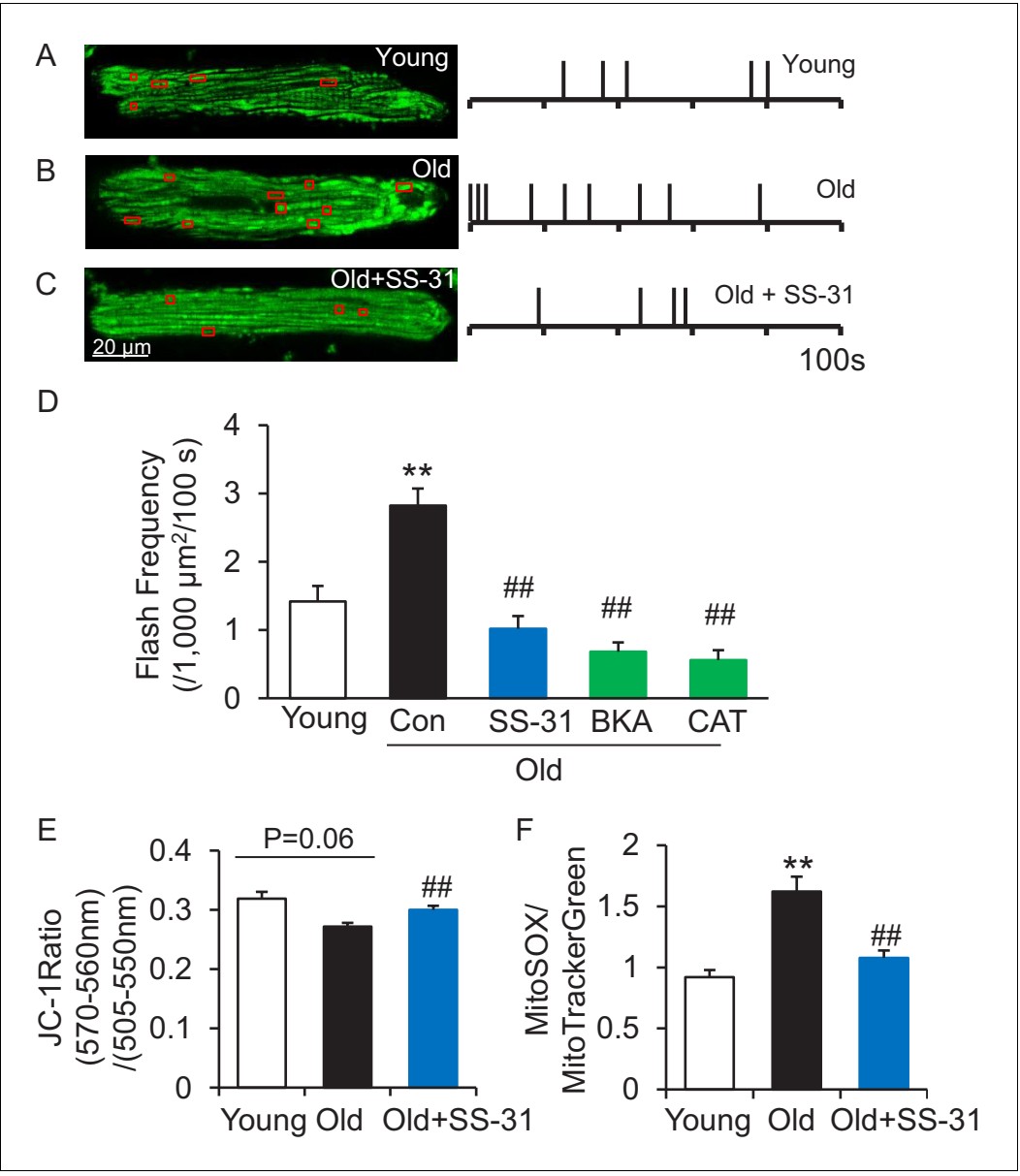

**Figure 4.** SS-31 attenuates excessive mitoflash activity in aged rat cardiomyocytes. (**A–C**) Mitoflash events within the regions shown in the red boxes took place at the times shown by vertical bars during the 100 s scanning time in the representative cardiomyocytes from young (**A**), old (**B**) and old+SS-31 (**C**) rat hearts. (**D**) The rate of mitoflash activity was increased in old rat cardiomyocytes compared to young, but 1 hr SS-31, Bongkrekic Acid (BKA) (10 μM) and Carboxyatractyloside (CAT) (20 μM) treatments decreased the mitoflash frequency in old cells to or below that of young cells. N = 26–87 cells from 3 to 14 rats. **p<0.05 vs young. ##p<0.05 vs old. (**E**) JC-1 red to green fluorescence ratio, indicative of mitochondrial membrane potential, in cells from young and old mice and old mouse cardiomyocytes treated with SS-31 (10 μM for 12 hr). N = 84–218 cells from 3 to 4 mice. p=0.06 vs young. ##p<0.01 vs old. (**F**) Mitochondrial ROS production in mouse cardiomyocytes measured by the fluorescence ratio of MitoSOX (5 μM, excitation 540 nm, emission >560 nm) to Mitotracker green (200 nM, excitation 488 nm, emission 505–530 nm). One-way ANOVA with Dunnett's test was applied. N = 40–84 cells from 3 to 5 mice. **p<0.05 vs young. ##p<0.05 vs old.

The online version of this article includes the following figure supplement(s) for figure 4:

**Figure supplement 1.** Increased mitochondrial flash activity in the intact perfused mouse aged heart.

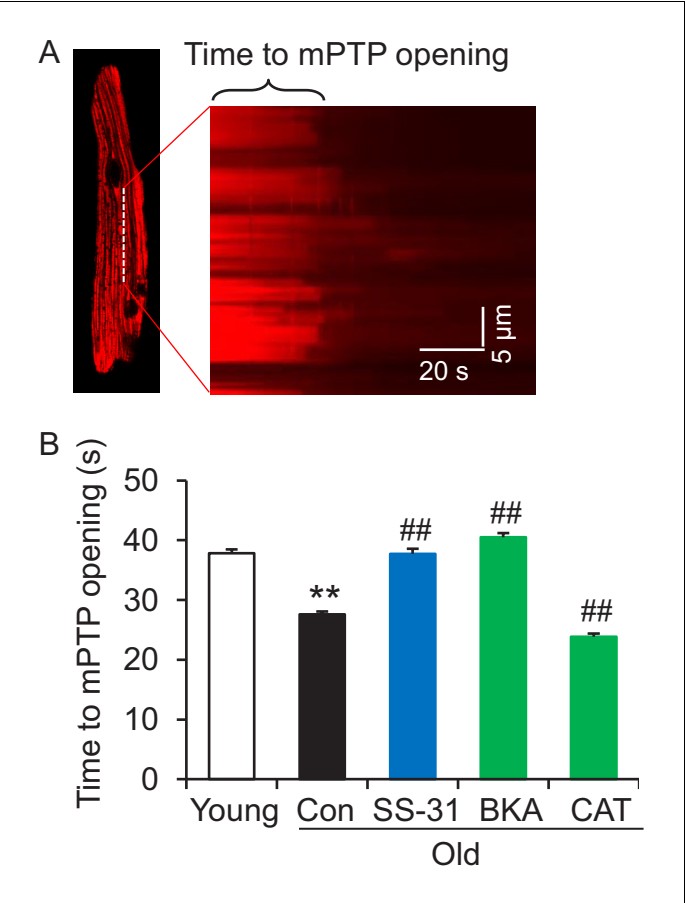

**Figure 5.** SS-31 reverses the increased speed of mitochondrial permeability transition pore (mPTP) opening in aged mouse cardiomyocytes. (**A**) A typical image shows 1 Hz line-scanning photo-excitation induced mPTP opening in a cardiomyocyte loaded with the mitochondrial membrane potential ($\Delta\psi_m$) dye Tetramethylrhodamine methyl ester (TMRM). The sudden decline of TMRM fluorescence with time (rightward) indicates mPTP opening and $\Delta\psi_m$ loss. (**B**) 1 hr SS-31 and Bongkrekic Acid (BKA), but not carboxyatractyloside (CAT) treatments protect the photo-excitation induced mPTP opening. Quantification of time to mPTP opening from 418 to 658 mitochondria from 19 to 32 cells isolated from 3 to 4 mice in each group. One-way ANOVA with Dunnett's test was applied. **$p<0.01$ vs young, ##$p<0.01$ vs old.

## Discussion

In this report, we have shown direct evidence of increased proton leak in the aged mitochondria as a primary energetic disturbance and that the increased proton entry in old cardiomyocytes likely takes place through ANT1. Moreover, we demonstrated that SS-31 prevents the proton entry to the mitochondrial matrix and rejuvenates mitochondrial function through direct interaction with ANT1 and stabilization of the ATP synthasome. During aging, the pathological augmented and sustained basal proton leak burdens the mitochondrial work load, resulting in a decline in respiratory efficiency. Blocking this pathological proton leak induced by aging benefits the mitochondria and the heart (*Figure 7*). We suggest that the restoration of aged mitochondrial function that is conferred by SS-31 is directly attributable to this effect. However, the resulting enhancement in diastolic function is likely to require downstream changes, as the functional benefit took up to 8 weeks to reach full effect, and required post-translational modifications of contractile protein elements (*Chiao et al., 2020*). It is increasingly recognized that mitochondrial function, including redox status and energetics, has far-reaching effects, including epigenetic alterations and post-translational modifications (*Olgar et al., 2019*).

ANT1 appears to mediate the pathological mitochondrial proton leak in the aged mouse heart. Although an increased mitochondrial proton leak in the aged heart was previously suggested by

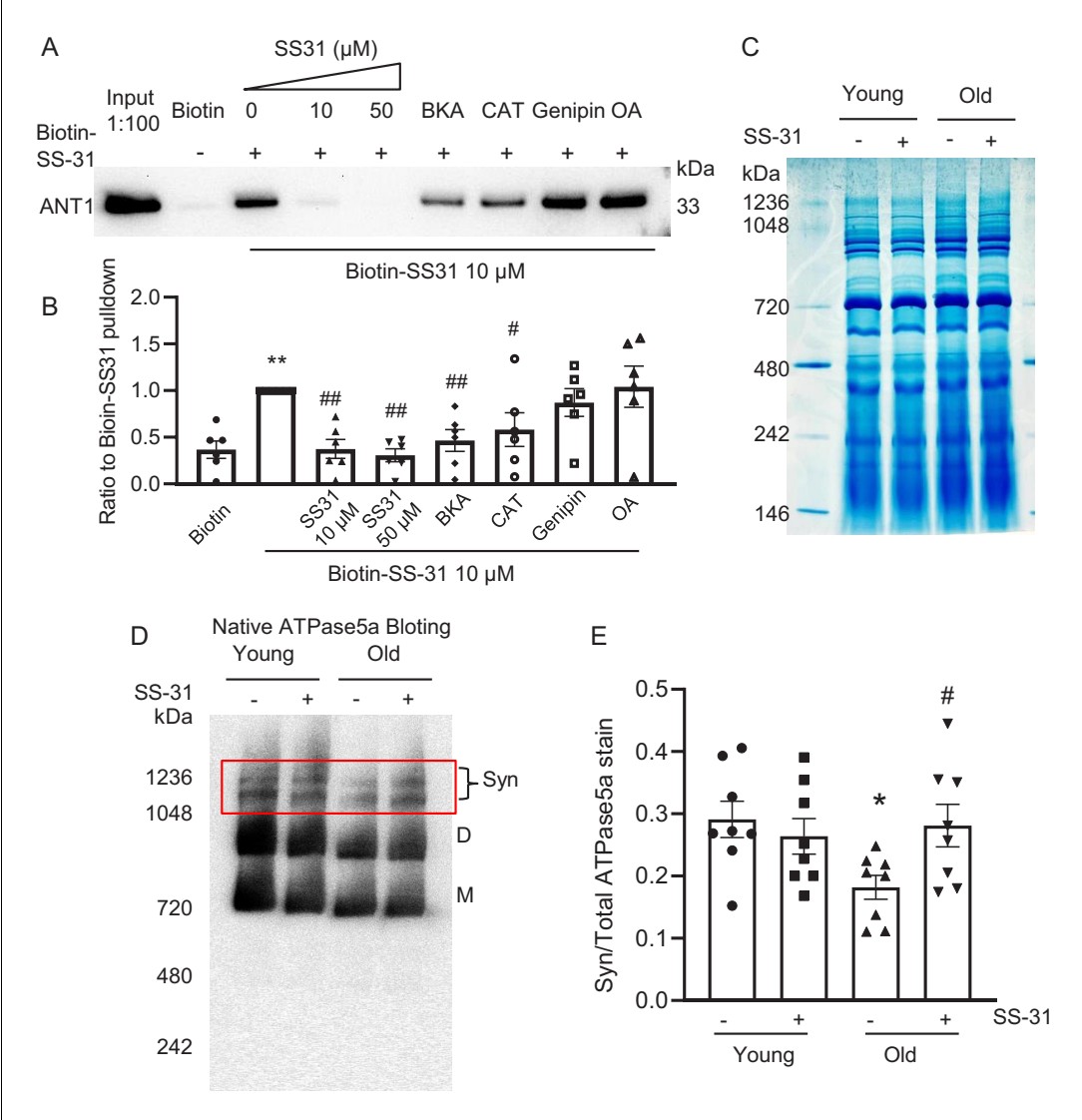

**Figure 6.** SS-31 interacts with ANT1 and stabilizes the ATP synthasome in the old mouse heart mitochondria. (**A, B**) Biotin-SS-31 pulldown shows the association of biotin-SS-31 to ANT1. Free SS-31 competes with this interaction, while Bongkrekic Acid (BKA) and carboxyatractyloside (CAT) inhibit the interaction of biotin-SS-31 with ANT1. Panel A shows a representative Western blot. N = 6 mice in each group. One-way ANOVA with Fisher's LSD test was applied. **p<0.01 vs Biotin control, #p<0.05, ##p<0.01 vs Biotin-SS31 pulldown. (**C**) Coomassie blue staining of isolated mitochondria in a native gel. (**D**) Left panel is the total protein loading control for the Native gel blot. (**D, E**) Native gel blotting shows that 10 μM SS-31 stabilizes the mitochondrial synthasome (Syn) in isolated mitochondria. The Syn is highlighted in the red box. The Syn and ATPase Dimer (**D**) and Monomer (**M**) were labeled using anti-ATP5A. N = 8 mice in each group. One-way ANOVA with Fisher's LSD test was applied. *p<0.05 vs young, #p<0.05 vs old. The online version of this article includes the following figure supplement(s) for figure 6:

**Figure supplement 1.** Silver staining of the Biotin-SS-31 pulldown of *Figure 6A* from the old mouse heart mitochondria.

indirect oxygen consumption measurement (*Serviddio et al., 2007*), the site of this augmented proton leak in aging mitochondria has remained a puzzle. We directly evaluated the proton leak using the mitochondrial matrix-targeted pH indicator (mt-cpYFP) and provide evidence that implicates ANT1, the ATP/ADP translocator, as responsible for the pathologically increased proton leak in aged cardiomyocytes. This does not necessarily implicate the ADP/ATP translocase mechanism itself in the proton leak, as in the unenergetic state in which we examined the mitochondrial pH resistance, there would be no ADP/ATP transport activity. Our result is, however, supported by a recent report that proton transport is an integral function of ANT1 (*Bertholet et al., 2019*). Because the ANT1 protein level is not increased in the aged heart (it is, in fact, mildly but significantly decreased,

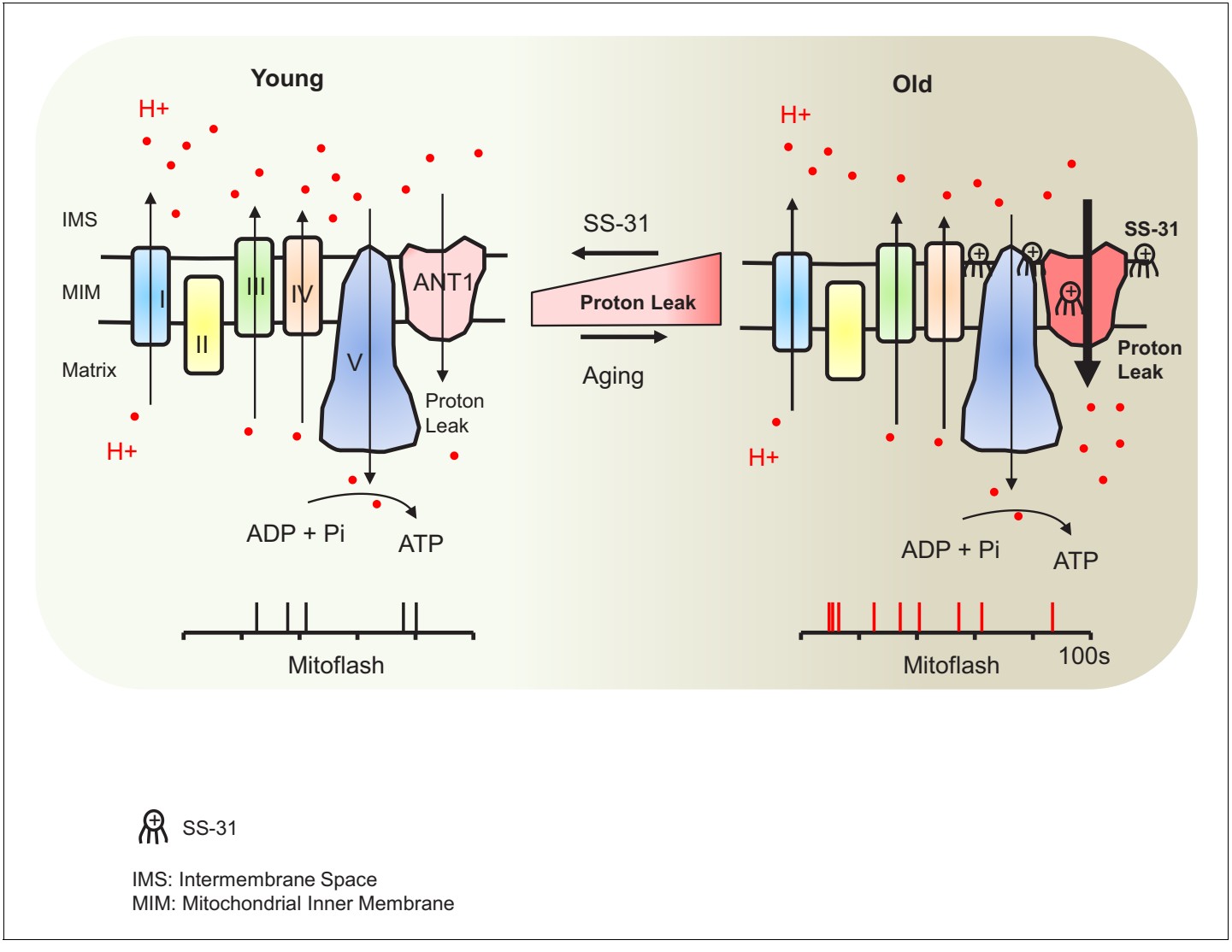

**Figure 7.** Schematic of the mechanism of SS-31 protection of proton leak and rejuvenation of mitochondrial function. Due to increased mitochondrial proton leak, the mitochondria work harder to maintain ATP production, and thus the work load is increased in the aged heart.

*Figure 3—figure supplement 1*), the aging-augmented proton leak through ANT1 must be through altered transport activity or conformational change. Both the inhibitors BKA (locking ANT1 in the matrix face state [*Ruprecht et al., 2019*]) and CAT (locking ANT1 in the cytosol face state [*Pebay-Peyroula et al., 2003*]) suppressed the proton leak in the aged cardiomyocytes, suggesting that constraining the conformational state in either position, or otherwise blocking the proton pore reduces ANT1 proton translocation.

Most interestingly, for the first time, we showed that a novel drug, SS-31 (elamipretide), now in clinical trials, prevents the augmented mitochondrial proton leak, rejuvenates mitochondrial function, and reverses aging-related cardiac dysfunction. Mechanistically, we found that SS-31 directly interacts with ANT1 and stabilizes formation of the ATP synthasome. This would seem surprising, given the prior belief that SS-31 affects mitochondria via binding to cardiolipin. However, the notion that SS-31 prevents the proton leak by direct interaction at the pore 'pocket' of ANT1 is supported by recent observations based on cross-linking 'interactome' mass-spectroscopy that showed that SS-31 is in intimate proximity to two lysine amino acid residues in the water filled cavity of the ANT1 protein (*Chavez et al., 2020*). Moreover, the cross-linking data suggested that this interaction could have structural consequences and may stabilize the m-state of ANT1 (*Chavez et al., 2020*). Our observation that both BKA and CAT blocked the SS-31 interaction with ANT1 suggests that SS-31

interacts with ANT1 independent of the ANT1 face to m-state (matrix facing) or c-state (cytoplasmic-facing). These results, and prior evidence of the critical role of ANT1 in mitochondrial health and function (*Liu and Chen, 2013*), warrant further high resolution structural study of the ANT transporter.

It has recently been shown that SS-31 alters surface electrostatic properties of the mitochondrial inner membrane (*Mitchell et al., 2019*). The consequences of this effect could include alteration of the channel ion gating properties of the ANT, including conformational changes secondary to enhanced supercomplex and ATP synthasome complex stability. SS-31 effects on stabilization of mitochondrial synthasome (*Figure 6D,E*) could directly contribute to the enhanced efficiency of mitochondrial respiration that is seen in muscle of SS-31 treated old animals (*Siegel et al., 2013*) and the improvement in performance of humans with primary mitochondrial myopathy (*Karaa et al., 2018*). As considerable prior literature indicates that SS-31 interacts with cardiolipin, this interaction would also affect the ATP synthasome, as all the components of the ATP synthasome are 'floating' in the inner membrane lipid membrane (of which about 20% is cardiolipin). It is possible that this directly affects stability of the synthasome components. Alternatively, the effect could be due to the change in inner membrane electrostatic properties that has recently been shown to result from incorporation of SS-31 into the inner membrane (*Mitchell et al., 2019*), as this could indirectly stabilize the ATP synthasome.

The restoration of membrane potential by SS-31 in the old mitochondria (*Figure 4E*) can be attributed to the suppression of proton leak. However, SS-31 also decreased ROS production (*Figure 4F*) in the aged cardiomyocytes. It is unclear if the reduced ROS production is associated with modification of the ANT1 shown here or through a parallel mechanism. Blocking this pathological proton leak induced by aging will benefit the mitochondria and the heart. This is not in conflict with the 'uncoupling to survive hypothesis', which arises from the positive correlations between increase proton leak, reduced ROS, and increased lifespan (*Brand, 2000*). This reduced ROS production is interpreted as resulting from decreased electromotive force and consequent reduced electron leak during transport through the respiratory chain. However, SS-31 through its interaction with cardiolipin, abundant in the inner membrane, can improve the efficiency of electron transfer, especially by its known interaction with the heme group of cytochrome c (*Szeto, 2014*; *Szeto and Birk, 2014*), thereby reducing ROS production, even as the aged mitochondrial membrane potential is increased. Thus, our data support the conclusion that SS-31 interaction with multiple inner membrane proteins enhances the performance of multiple facets of respiratory mechanics.

In summary, our study reveals that the adenine nucleotide transporter is the most likely candidate responsible for the elevated proton leak in old cardiomyocytes and that SS-31 acutely and directly interacts with the transporter, preventing the proton leak and rejuvenating mitochondrial function in the aged cardiomyocytes. The improved mitochondrial function leads to complex secondary changes to effect enhanced diastolic function in the aged heart. These findings provide a novel insight for better understanding of the mechanisms of cardiac aging and establish the novel concept that decreasing the pathological proton leak in the aging heart restores mitochondrial function, ultimately reversing cardiac dysfunction in aging.

# Materials and methods

**Key resources table**

| Reagent type (species) or resource | Designation | Source or reference | Identifiers | Additional information |
|---|---|---|---|---|
| Strain, strain background (*M. musculus*; male and female) | C57BL/6J | National Institute of Aging Charles River colony | RRID:IMSR_JAX:000664 | See Materials and method |
| Strain, strain background (*Rattus norvegicus*) | F344 | National Institute of Aging F344 rats | | See Materials and method |
| Genetic reagent (*M. musculus*) | mt-cpYFP | Wang lab, PMID:25252178 | | See Materials and method |

*Continued on next page*

*Continued*

| Reagent type (species) or resource | Designation | Source or reference | Identifiers | Additional information |
|---|---|---|---|---|
| Transfected construct (Adenovirus) | mt-cpYFP | Wang lab, PMID:25252178 | | Adenovirus to transfect and express mt-cpYFP |
| Antibody | anti- ANT1 | Abcam | Cat# ab102032 | 1:3000 |
| Antibody | anti-UCP2 | Cell signaling technology | Cat# 89326S | 1:2000 |
| Antibody | anti-ATP5a | Abcam | Cat# ab14748 | 1:3000 |
| Peptide, recombinant protein | SS-31 peptide (Elamipretide) | Stealth Bio Therapeutics | | |
| Commercial assay or kit | Seahorse XF Cell Mito Stress Test Kit | Aligent/Seahorse Bioscience | 103015–100 | |
| Commercial assay or kit | MitoSOX Red | ThermoFisher | M36008 | |
| Chemical compound, drug | Bongkrekic acid | Cayman Chemical | 19079 | |
| Chemical compound, drug | Carboxyatractyloside | Cayman Chemical | 21120 | |
| Chemical compound, drug | Genipin | Sigma | G4796 | |
| Chemical compound, drug | Oligomycin A | Sigma | 75351 | |
| Chemical compound, drug | carbonyl cyanide-p-trifluoromethoxyphe nylhydrazone | Sigma | C2920 | |
| Chemical compound, drug | Antimycin A | Sigma | A8674 | |
| Chemical compound, drug | Rotenone | Sigma | R8875 | |
| Commercial assay or kit | MitoTracker Green | ThermoFisher | M7514 | |
| Commercial assay or kit | JC-1 Dye | ThermoFisher | T3168 | |
| Commercial assay or kit | Biotin | ThermoFisher | B20656 | |
| Commercial assay or kit | Streptavidin Agarose beads | ThermoFisher | 20349 | |
| Commercial assay or kit | Pierce Reversible Protein Stain Kit for PVDF Membranes | Thermo Scientific | 24585 | |
| Commercial assay or kit | BCA protein assay | Thermo Scientific | 23225 | |
| Commercial assay or kit | SuperSignal West Pico PLUS Chemilu minescent Substrate | Thermo Scientific | 34580 | |
| Software, algorithm | Graphpad Prism | Graphpad | RRID:SCR_002798 | |
| Software, algorithm | AlphaView Software | ProteinSimple | | |

## Animals

All the animal procedures were approved by the Institutional Animal Care and Use Committee at the University of Washington and conform to the NIH guidelines (Guide for the care and use of laboratory animals). The use of mice and rats have been approved by IACUC under animal protocols 2174–23 and 2654–03. Young (4–6-month-old) and aged (24–26-month-old) C57BL/6 mice (Charles River colony) and F344 rats (young, 5-7-month old; aged, 25–30-month-old) were obtained from the

National Institute of Aging Rodent Resource. The mt-cpYFP transgenic C57BL/6 mice were housed until reaching the age described.

## Isolation of adult mouse and rat cardiomyocytes

Single ventricular myocytes were enzymatically isolated from mouse and rat hearts as described previously (*Zhang et al., 2013*; *Zhang et al., 2017*). The rod-shaped cardiomyocytes were collected by allowing cells to settle down and adhere to laminin coated-24 well Seahorse plates for intact cell oxygen consumption test, or to glass coverslips for confocal imaging.

## Seahorse assay

The XF24e Extracellular Flux Analyzer (Seahorse Bioscience) was used for measuring oxygen consumption in intact resting cardiomyocyte, with XF assay medium containing 5 mM glucose and 1 mM pyruvate. Oligomycin A (OA, 2.5 μM), carbonyl cyanide-p-trifluoromethoxyphenylhydrazone (FCCP, 1 μM), and antimycin A (AA, 2.5 μM) plus 1 μM rotenone (Rot) were added in three sequential injections, as reported (*Zhang et al., 2017*).

## Confocal imaging

We used a Zeiss 510 (Zeiss, Germany) or Leica SP8 (Leica, Germany) for confocal imaging at room temperature. The cells were placed in modified Tyrode's solution (in mM: 138 NaCl, 0.5 KCl, 20 HEPES, 1.2 MgSO$_4$, 1.2 KH$_2$PO$_4$, 5 Glucose, 1 CaCl$_2$, pH 7.4). For mitochondrial flashes, mt-cpYFP expressing cells were exposed to alternating excitation at 405 and 488 nm and emission collected at >505 nm. Time-lapse 2D images were collected at a rate of 1 second per frame. For mitochondrial superoxide quantitation, we used the ratio of MitoSOX Red (5 μM, excited at 540 nm with emission collected at >560 nm) to mitoTracker Green (200 nM, excited at 488 nm and emission collected at 505–530 nm). For mitochondrial membrane potential measurement, JC-1 was excited at 488 nm and emission collected at 510–545 nm and 570–650 nm. For photon triggered mPTP opening, the cells were loaded with 120 nM Tetramethylrhodamine methyl ester (TMRM) and line scanned at 1 Hz as described previously (*Zorov et al., 2000*).

## Cell permeabilization and pH stress

Rat cardiomyocytes were cultured with mt-cpYFP adenovirus (*Wang et al., 2016b*) for 3 days in M199 medium. After incubation in Ca$^{2+}$-free Tyrode's solution for 30 min, the medium was changed to a solution of 100 mM potassium aspartate, 20 mM KCl, 10 mM glutathione, 10 mM KH$_2$PO$_4$, 0.1 mM EGTA, 8% dextran 40,000, pH 7.5, with 50 μg/ml saponin for 30 s and then maintained in saponin-free internal solution (*Lukyanenko and Gyorke, 1999*). The pH of the solution containing the permeabilized cells was then progressively lowered by addition of HCl in quantities previously titrated to result in pH 7.3, 6.9, 5.3, and 4.5, with 8 min between each step. The permeabilized cells were excited using the same settings as for mt-cpYFP above, but using a time-lapse of 6 seconds per frame. The ratio of emission fluorescence at 488 nm from 405 nm excitation indicated the mitochondrial pH change (*Wei-LaPierre et al., 2013*) and was normalized to a starting (pH 7.5) arbitrary value of 1.0, so as to normalize differences due to variability of the intensity of laser excitation and emission collection between different experiments. Bongkrekic acid (BKA) 10 μM (*Adams et al., 2000*), CAT 20 μM (*Winter et al., 2016*), Genipin 50 μM (*Wang et al., 2017*), OA were added after cell permeabilization.

## Western blots

Heart tissue was lysed with RIPA buffer containing a protease inhibitor cocktail (*Chiao et al., 2016*). Protein samples were denatured and separated via NuPAGE Bis-Tris gel, and transferred to PVDF membranes. The blots were probed with primary antibodies: ANT1 (Abcam, ab102032, 1:3000), UCP2 (Cell Signaling Technology, 89326S, 1:2000) followed by appropriate secondary antibodies.

## Biotin-SS-31 pulldown and blot analysis

Hearts were chunked and dounce homogenized in mitochondrial isolation buffer (MIB, in mM: 300 sucrose, 10 Na-HEPES, 0.200 EDTA, pH 7.4) and centrifuged at 800 g for 10 min. The supernatants were centrifuged at 8000 g for 15 min to purify mitochondria. Digitonin was added to the

mitochondria at a ratio of Digitonin: protein = 6:1 to break down the membrane system. Treatment drugs were added 30 min before addition of 10 µM biotin-SS-31 (Biotin-D-Arg-dimethyl Tyr-Lys-Phe-NH2) or biotin control (Thermo, B20656). Streptavidin Agarose beads (Thermo, 20349) were added and incubated for 2 hrs at room temperature. The beads were washed with MIB three times and then eluted by 50 µM SS-31. The eluates were boiled with LDS protein loading buffer (Thermo, NP0008) and loaded on NuPAGE for gel electrophoresis and Western blotting with antibody for ANT1 (Abcam, ab102032, 1:3000). In some experiments, after electrophoresis, gels were silver stained using a Pierce Silver Stain Kit (Thermo, #24612).

## Native coomassie blue staining and blotting

Mitochondria from mouse hearts were isolated as described previously (*Marcu et al., 2012*). Mitochondria (100 µg) were solubilized in 4x NativePAGE Sample Buffer containing 5% digitonin and 5% coomassie blue G-250. The samples were loaded on NativePAGE Novex 3–12% Gel and run at 100 V for 1 hr, then at 300 V for 2 hr. For coomassie blue staining, gels were stained with 0.1% Coomassie Brilliant Blue overnight and destained with destaining solution ($H_2O$: Methanol: Acetic Acid = 5:4:1) 5 times at 20 min intervals. For native blotting, gels were transferred to PVDF membranes at 25 V in 4°C overnight and incubated with ATP5a antibody (Abcam, ab14748, 1:3000), followed by anti-mouse secondary antibody.

## Perfused mouse heart confocal imaging

mt-cpYFP transgenic mice were anesthetized with pentobarbital (150 mg/kg). The heart was removed, cannulated via the ascending aorta, and put on a modified perfusion system and in a custom made chamber on the confocal stage as previously reported (*Zhang et al., 2018*; *Zhang et al., 2017*). The perfusion was maintained under a constant flow (~2 ml/min) with $O_2/CO_2$-bubbled KHB solution (in mM: 118 NaCl, 0.5 EDTA, 10 D-glucose, 5.3 KCl, 1.2 $MgCl_2$, 25 $NaHCO_3$, 0.5 pyruvate, and 2 $CaCl_2$, pH 7.4) at 37°C. To minimize motion artifact during imaging, 10 µM (-)-Blebbistatin (Toronto Research Chemicals) was included. During imaging, the left ventricle was gently pressed to further suppress motion artifact. Mitoflashes were imaged using the procedure described above.

## Data statistics

Data are shown as mean ± SEM. For the multiple groups comparisons one-way ANOVA was applied followed with appropriate test. Student's t-test was used to determine the statistical significance between two group comparisons. $p < 0.05$ was considered statistically significant.

## Acknowledgements

We thank Drs. Mariya Sweetwyne, Ying Ann Chiao, Martin Brand, Michael MacCoss and Gaomin Feng for technical support and helpful discussions and the services of the WM Keck Microscopy Center at the University of Washington. SS-31 (elamipretide) was kindly provided by Stealth Biotherapeutics (Newton, MA).

This work was supported by NIA P01AG001751 and R56AG055114 to PSR, HL114760, HL137266 and AHA 18EIA33900041 to WW, and a Glenn Foundation for Medical Research Postdoctoral Fellowship and AHA 19CDA34660311 to HZ.

## Additional information

### Competing interests

Hazel Szeto: is the inventor of SS-31 and founder of Stealth Biotherapeutics. The other authors declare that no competing interests exist.

### Funding

| Funder | Grant reference number | Author |
| --- | --- | --- |
| NIA | P01AG001751 | Peter S Rabinovitch |

| | | |
|---|---|---|
| NIA | R56AG055114 | Peter S Rabinovitch |
| NHLBI | HL114760 | Wang Wang |
| NHLBI | HL137266 | Wang Wang |
| AHA | 18EIA33900041 | Wang Wang |
| AHA | 19CDA34660311 | Huiliang Zhang |
| Glenn/AFAR Foundation | Medical Research Postdoctoral Fellowship | Huiliang Zhang |

The funders had no role in study design, data collection and interpretation, or the decision to submit the work for publication.

### Author contributions

Huiliang Zhang, Conceptualization, Resources, Data curation, Formal analysis, Funding acquisition, Investigation, Visualization, Methodology, Writing - original draft; Nathan N Alder, Conceptualization, Resources, Methodology, Writing - review and editing; Wang Wang, Resources, Funding acquisition, Writing - review and editing; Hazel Szeto, Writing - review and editing; David J Marcinek, Resources, Writing - review and editing; Peter S Rabinovitch, Conceptualization, Resources, Funding acquisition, Writing - original draft, Project administration

### Author ORCIDs

Huiliang Zhang ⓘ https://orcid.org/0000-0002-8967-1219
Peter S Rabinovitch ⓘ https://orcid.org/0000-0001-7169-3543

### Ethics

Animal experimentation: All the animal procedures were approved by the Institutional Animal Care and Use Committee at the University of Washington and conform to the NIH guidelines (Guide for the care and use of laboratory animals). The use of mice and rats have been approved by IACUC under animal protocols 2174-23 and 2654-03.

### Decision letter and Author response

Decision letter https://doi.org/10.7554/eLife.60827.sa1
Author response https://doi.org/10.7554/eLife.60827.sa2

## Additional files

### Supplementary files

• Transparent reporting form

### Data availability

Data available in Dryad: https://doi.org/10.5061/dryad.fqz612jqs.

The following dataset was generated:

| Author(s) | Year | Dataset title | Dataset URL | Database and Identifier |
|---|---|---|---|---|
| Zhang H, Alder NN, Wang W, Hazel SH, Marcinek DJ, Rabinovitch PS | 2020 | Reduction of Elevated Proton Leak Rejuvenates Mitochondria in the Aged Cardiomyocyte | https://doi.org/10.5061/dryad.fqz612jqs | Dryad Digital Repository, 10.5061/dryad.fqz612jqs |

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
