## [Decision Letter]

**Acceptance summary:**

Your work is novel and a clear demonstration of the potential role of mitochondrial proton leak in cardiac aging and dysfunction and the protective effect of SS-31.

**Decision letter after peer review:**

Thank you for submitting your article "Reduction of Elevated Proton Leak Rejuvenates Mitochondria in the Aged Cardiomyocyte" for consideration by *eLife*. Your article has been reviewed by three peer reviewers, and the evaluation has been overseen by a Reviewing Editor and Jessica Tyler as the Senior Editor. The following individual involved in review of your submission has agreed to reveal their identity: Benjamin Miller (Reviewer #2).

The reviewers have discussed the reviews with one another and the Reviewing Editor has drafted this decision to help you prepare a revised submission.

In this manuscript Zhang et al. examine the effects of the compound SS-31, a compound that has shown intervention potential in other systems, on increased mitochondrial proton leak in cardiac myocytes from aged rats and mice. The data are important contribution to the mechanism of action of SS-31, which shows great promise for cardiac aging and the interaction of SS-31 with ANT1. ANT1 is the major isoform of ANT expressed in the heart, and it has been shown previously to contribute to proton leak in cardiac mitochondria. The authors also analyze the effects of SS-31 on mito-flash and mitochondrial permeability transition pore opening. From the work presented here the authors conclude that proton leak is an important underlying mechanism of cardiac dysfunction in aging and provide evidence for the effect of SS-31 in modulating this effect.

While the work was deemed interesting, however the manuscript should be improved by more clarity in the presentation of the data and their interpretation in general. There are several significant concerns that need to be addressed:

Essential revisions:

1) This work does not appear to be a major advance beyond the author's recent publication in *eLife* showing that "SS-31 normalized the increase in proton leak and reduced mitochondrial ROS in cardiomyocytes from old mice, accompanied by reduced protein oxidation and a shift towards a more reduced protein thiol redox state in old hearts." They also demonstrated improved diastolic function. (Chiao et al., 2020).

2) Authors cannot claim that they have "identified ANT1 as mediating the increased proton permeability of old cardiomyocytes". ANT2 is also expressed, albeit at lower levels than ANT1, in the heart, and is also inhibited by CATR and BKA (e.g., work from Doug Wallace's lab: PMID: 10974536) – and thus could also contribute to the effects of SS-31. Authors state that Bertholet et al., 2019, found that ANT1 was the major site of proton leak; while this is true, they also found that ANT2 can contribute to leak, especially FA- induced leak. Thus authors cannot claim, as they have in the their summary statement that their "study reveals that ANT1 is responsible for the elevated proton leak in old cardiomyocytes and that SS-31 directly interacts with ANT1, preventing the proton leak and rejuvenating mitochondrial function in the aged cardiomyocytes." Indeed, data in Figure 1 show that SS-31 inhibition of proton leak is significantly less than the inhibition caused by BKA; BKA also inhibits ANT2.

3) Moreover there are other mechanisms contributing to proton leak that could be relevant here: NNT, UCP3 and the inorganic phosphate carrier. See for example the recent work on NNT and its contributions to leak and ROS (Smith et al. JBC 2020; PMID 32747443). Why have the authors not shown blots of ANT2 and other possible proteins involved? Regarding UCP2 blots, authors should show a positive control (e.g., a recombinant protein) in the western blots to convince readers that this band really is UCP2. The antibodies for UCP2 are very poor, and UCP2 protein expression is extremely low, if detectable at all in the heart.

4) We felt that the authors over-interpreted their mt-cp-YFP results. Authors conducted these analyses in the absence of mitochondrial activity, and permeabilized the cardiomyocytes in a buffer that contained no substrates, ATP, or ADP. If authors had found effects under physiological conditions, this would be meaningful. Moreover, the control of the pH of the mitochondrial matrix is not via mitochondrial proton leak; there are many other ion transport mechanisms in the mitochondrial inner membrane and there are many mechanisms in the matrix itself. Authors need to conduct analyses under physiological conditions in the presence of energy substrates.

5) Authors describe how oligomycin failed to inhibit proton leak, but it is well known that it does not inhibit proton leak. Authors need to revise this section.

6) Authors use the well-known inhibitors of ANT, ie. CAT and BKA in non-permeabilized cells, and it is generally thought that they are not cell permeable (PMID: 26950698). Authors need to demonstrate that the inhibitors are traversing the cell membrane.

7) Seahorse data for the single ventricular myocytes of rats and mice are expressed in units per 800 cells. How did authors assess the number of cells that actually adhered and thus that were actually measured in these assays?

8) Why is MitoSOX normalized to mitochondrial content? Are there differences in mitochondrial content in the different experimental groups (old vs young)?

9) Models are important for interpretation. The initial experiment specifies primary cardiac myocytes whereas subsequent studies indicate cardiac myocytes. I think it is important to know for sure what model was used.

10) There are multiple statements of "rejuvenation" "restoring" "more youthful" etc. In a strict sense, the data compare young to old and to old+SS-31. These groups are similar or different. It is not possible to say that the a given group of cells changed with age and with treatment were made like their young self again as implied by the use of these words.

11) How do the authors justify using t-tests? Within each experiment were multiple comparisons so t-tests do not seem justified.

12) The presentation of the figures bounces around quite a bit and are not in sequence. This makes the study hard to follow. In addition, the figures lack detail such as what the abbreviations are and what the error bars are. Although these seem like minor quibbles, when added up, it made it difficult to assess the data.

13) Why was normalization used in many of the assays? Was there a lot of variability at baseline? And if so, why?

14) What was the justification for carrying out experiments in Figure 2 at pH 5.3? This pH does not seem relevant for general mitochondrial function. Do you see the same results at a pH close to actual mitochondria pH?

15) The Discussion made some conclusions that were maybe a bit too much speculation. For example, the proton leak is identified as pathological, although this was not directly demonstrated in the study. Also, protein conformation was inferred as well.

16) It should be emphasized that the proton leak studied is in myocytes in the resting, non-working state oxidizing endogenous substrates. The lack of a resting defect in OCR in this state is therefore plausible compared to the extensive literature describing key defects in OXPHOS with aging.

17) The myocyte yield of the isolation procedure should be discussed. Are only the "best" aged myocytes obtained?

18) The pH dependence of 405 excitation is of some concern. Also, calibration curves should be shown for pH 5.3 and 6.9; which are critical to the work.

19) The proton leak shown is of interest since state 4 rates in isolated cardiac mitochondria have minimal change with age. This should be discussed and reconciled.

20) The similarity of BKA and CAT to block proton leak is somewhat puzzling since the response of MPTP to the inhibitors is of course different. Is there evidence of MPTP opening in the permeabilized cells used for the measurement of mitochondrial pH with the buffer used as the artificial "cytosol"?

21) The differences in Figure 2A are modest; the "normalized" data in Figure 2B and Figure 3C are more reasonable. How were fluorescence data normalized?

22) The dynamic range and reproducibility of the cp-YFP assay should be described in greater detail. It seems puzzling that some groups apparently required n=19 for valid results.

23) Is there a quality control assessment of mitochondrial inner membrane integrity in the permeabilized cells at low pH? This is critical to the conclusion of the study.

---

## [Author Response]

Essential revisions:1) This work does not appear to be a major advance beyond the author's recent publication in eLife showing that "SS-31 normalized the increase in proton leak and reduced mitochondrial ROS in cardiomyocytes from old mice, accompanied by reduced protein oxidation and a shift towards a more reduced protein thiol redox state in old hearts." They also demonstrated improved diastolic function. (Chiao et al., 2020).

We wish to respectfully disagree with this interpretation, as the difference between a functional observation (prior work) and elucidation of mechanism (present work) is large. In the recent publication in *eLife* we showed that chronic SS-31 8-week osmotic minipump treatment decreased the proton leak in aged mice and improved diastolic function. However, the mechanism for this effect was unclear; in fact, the long-term in vivo treatment could even have been produced by indirect compensatory changes, including proteomic or metabolic remodeling, as opposed to a possible direct effect on the cardiomyocyte. Moreover, the underlying mechanism of the excessive proton leak remained elusive. The present study showed that SS-31 directly and acutely decrease excessive proton leak; by directly evaluating the physical properties of the inner membrane in the absence of any active metabolic process we were able to determine the mechanism of the increased proton leak and how SS-31 prevents it in the aged mitochondria. We specified the novelties in the revision as “SS-31 acutely prevents the excessive mitochondrial proton entry”; “In this report we have shown direct evidence of increased proton leak in the aged mitochondria as a primary energetic disturbance and evidence that the increased proton entry in old cardiomyocytes takes place likely through ANT1. Moreover, we demonstrated that SS-31 prevents the proton entry to the mitochondrial matrix and rejuvenates mitochondrial function through direct interaction with the adenine nucleotide transporter and stabilization of the ATP synthasome.”; and “our study reveals that the adenine nucleotide transporter is the most likely candidate responsible for the elevated proton leak in old cardiomyocytes and that SS-31 acutely and directly interacts with the transporter, preventing the proton leak and rejuvenating mitochondrial function in the aged cardiomyocytes”.

2) Authors cannot claim that they have "identified ANT1 as mediating the increased proton permeability of old cardiomyocytes". ANT2 is also expressed, albeit at lower levels than ANT1, in the heart, and is also inhibited by CATR and BKA (e.g., work from Doug Wallace's lab: PMID: 10974536) – and thus could also contribute to the effects of SS-31. Authors state that Bertholet et al., 2019, found that ANT1 was the major site of proton leak; while this is true, they also found that ANT2 can contribute to leak, especially FA- induced leak. Thus authors cannot claim, as they have in the their summary statement that their "study reveals that ANT1 is responsible for the elevated proton leak in old cardiomyocytes and that SS-31 directly interacts with ANT1, preventing the proton leak and rejuvenating mitochondrial function in the aged cardiomyocytes." Indeed, data in Figure 1 show that SS-31 inhibition of proton leak is significantly less than the inhibition caused by BKA; BKA also inhibits ANT2.

Together with points raised in item #3 below, we agree that the possibilities of other sites as the proton leak contributor, including ANT2, should be considered. However, of these, the crosslinking interactome data recently published by our collaborators identified only ANT1 as interacting with SS-31 (PMID: 32554501). Thus, we are more confident to say ANT1 is involved in the proton leak in the aged mitochondria. As we agree that we cannot with certainty rule out the others sites of proton entry, we have revised the Abstract “our data suggested ANT1 as the most likely site of mediating increased mitochondrial proton permeability in old cardiomyocytes” and the Discussion to state that “our study reveals that the adenine nucleotide transporter is the most likely candidate responsible for the elevated proton leak in old cardiomyocytes”.

3) Moreover there are other mechanisms contributing to proton leak that could be relevant here: NNT, UCP3 and the inorganic phosphate carrier. See for example the recent work on NNT and its contributions to leak and ROS (Smith et al. JBC 2020; PMID 32747443). Why have the authors not shown blots of ANT2 and other possible proteins involved? Regarding UCP2 blots, authors should show a positive control (e.g., a recombinant protein) in the western blots to convince readers that this band really is UCP2. The antibodies for UCP2 are very poor, and UCP2 protein expression is extremely low, if detectable at all in the heart.

Specific to inorganic phosphate carrier, we checked the potential involvement of the mitochondrial phosphate carrier (PiC) in proton entry by applying the pH stress to mitochondria in a buffer without phosphate. The proton entry characteristics of mitochondria from old cells was indistinguishable from that of mitochondria in buffer containing phosphate. This data is added as Figure 3—figure supplement 2, with the statement in the text “Considering that the inorganic phosphate carrier might be a potential source of the elevated proton entry in old mitochondria, we examined the proton permeability of old mitochondria to pH stress in a buffer that did not contain phosphate; in this buffer, the proton entry characteristics of mitochondria from old cells was indistinguishable from that of mitochondria stressed in the presence of inorganic phosphate (Figure 3—figure supplement 2).”

With respect to UCP2, we confirmed the UCP2 WB in the heart samples with a UCP2 standard. The band we detected was confirmed as UCP2. This blot confirms the results now included as Figure 3—figure supplement 1E and referred to in the text. Moreover, our data is consistent with previous work from Sreekumaran Nair lab demonstrated that the lack of age-related change in UCP2 transcript levels (PMID: 11171595, see Figure 3 from that paper).

4) We felt that the authors over-interpreted their mt-cp-YFP results. Authors conducted these analyses in the absence of mitochondrial activity, and permeabilized the cardiomyocytes in a buffer that contained no substrates, ATP, or ADP. If authors had found effects under physiological conditions, this would be meaningful. Moreover, the control of the pH of the mitochondrial matrix is not via mitochondrial proton leak; there are many other ion transport mechanisms in the mitochondrial inner membrane and there are many mechanisms in the matrix itself. Authors need to conduct analyses under physiological conditions in the presence of energy substrates.

We strongly believe that performing the pH stress experiments in the absence of energy substrates is the clearest and most informative protocol. In the presence of energy substrates, the mitochondria will actively pump protons out of the matrix, making it more difficult and complicated to evaluate proton entry. Thus, the buffer without substrates allows us to evaluate the fundamental process of proton entry to the mitochondrial matrix. In this fashion, looking at the physical properties of the inner membrane in the absence of metabolic activity is highly informative and reduces many possible complications. Also, we do not claim that proton leak is the primary method of regulating pH, only that the age-related excessive leak is via ANT and is prevented by SS-31, and that this is correlated with improved behavior of MPTP opening, membrane potential and mitochondrial flashes in aged cardiomyocytes.

5) Authors describe how oligomycin failed to inhibit proton leak, but it is well known that it does not inhibit proton leak. Authors need to revise this section.

Thank you. We changed the description to “As expected, the ATPase inhibitor Oligomycin A failed to inhibit the proton leak”

6) Authors use the well-known inhibitors of ANT, ie. CAT and BKA in non-permeabilized cells, and it is generally thought that they are not cell permeable (PMID: 26950698). Authors need to demonstrate that the inhibitors are traversing the cell membrane.

We believe that this question is based on a misunderstanding – in the assay used in this report the plasma membrane is permeabilized by saponin. However, it may be of interest to point out that in cardiomyocytes both BKA (PMID: 19452617; PMID: 11110776; PMID: 26746144) and CAT (PMID: 26746144; PMID: 26548633) have been shown to be effective in non-permeabilized cells. This is indeed in contrast to the report cited by the reviewer (PMID: 26950698) which found that BKA and CAT had a milder effect on intact T98G cells than permeabilized cells.

7) Seahorse data for the single ventricular myocytes of rats and mice are expressed in units per 800 cells. How did authors assess the number of cells that actually adhered and thus that were actually measured in these assays?

For the cell number determination, we counted the cell numbers using hemocytometer after cell isolation. Only the rod shaped cells were counted as live cells. Usually the starting cell concentration was ~4*10^4^/ml. Then the cells were diluted to 8000 cell/ml and 100ul (containing ~800 cells) were plated in each Seahorse well. We had previously coated the plate wells with laminin (50ug/ml) to allow the cells to adhere to the bottom of the plate. The cells were allowed to attach to the plate for at least 2 hours before changed the culture medium to Seahorse Assay medium without disturbing the cells. In initial experiments Seahorse wells were examined by microscopy to confirm plated numbers and equality between young and old cardiomyocytes. As expected, no differences were seen between young and old cells. This time-consuming secondary confirmation was not continued in later experiments.

8) Why is MitoSOX normalized to mitochondrial content? Are there differences in mitochondrial content in the different experimental groups (old vs young)?

The reason that the ratio of MitoSOX to MitotrackerGreen is analyzed is to reduce the cell to cell variability in analysis, which is primarily due to differences in cell size and dye uptake between cells (cell variation in dye transport being largely the same for the two indicators allows the MitoTracker Green intensity to be used to normalize the changes seen in MitoSOX staining to make them independent of cell size and dye uptake). This ratiometric approach is exceedingly common in flow and image cytometry. There were no significant differences seen in average Mitotracker Green intensity between old vs young cells.

9) Models are important for interpretation. The initial experiment specifies primary cardiac myocytes whereas subsequent studies indicate cardiac myocytes. I think it is important to know for sure what model was used.

We appreciate this reminder. All experiments were performed with primary cardiac myocytes. For pH stress and mitochondrial flash experiments, rat cardiomyocytes were used; mouse heart and cardiomyocytes were used in all other experiments. We have revised the language used in the manuscript to clearly note this accordingly.

10) There are multiple statements of "rejuvenation" "restoring" "more youthful" etc. In a strict sense, the data compare young to old and to old+SS-31. These groups are similar or different. It is not possible to say that the a given group of cells changed with age and with treatment were made like their young self again as implied by the use of these words.

We are grateful for bringing this to our attention. In the revised manuscript we are careful to state only that the phenotypes of the treated cells are closer to that of young cells.

11) How do the authors justify using t-tests? Within each experiment were multiple comparisons so t-tests do not seem justified.

Thank you for bringing this to our attention. We did the t-tests as analysis of differences between the young vs. old and between old vs. a single treatment. In this revision, we applied one-way ANOVA analysis to all multiple group comparisons then followed this with student t-tests to look at pairwise comparisons, allowing readers to see both results. We updated the statistical analysis in the Materials and methods part in this revision.

12) The presentation of the figures bounces around quite a bit and are not in sequence. This makes the study hard to follow. In addition, the figures lack detail such as what the abbreviations are and what the error bars are. Although these seem like minor quibbles, when added up, it made it difficult to assess the data.

Thank you for this observation, which we have now corrected. The Figure 3C which showed the method of 488/405 ratio slope calculation was previously placed after the slope statistical analysis of in Figure 2D, which was hard to follow. We moved Figure 3C to become new Figure 2—figure supplement 3 and mentioned this before the statement of Figure 2D to make it easier to follow. Also, we added the full spellings for the abbreviations. All the error bars are standard errors of the mean and this information is added to the figure legends.

13) Why was normalization used in many of the assays? Was there a lot of variability at baseline? And if so, why?

Thank you for this question. The reasons for each normalization are as following:

For the pH stress assay, we normalized the cpYFP 488/405 ratio to the starting value. This is required due to inter-experimental variation in relative intensities of the two wavelengths due to variability in confocal laser powers, detector settings and efficiency.

For the superoxide evaluation, we normalized the mitoSOX signal to the MitotrackerGreen signal. The advantage of this normalization is described in response to query number 8, above.

For the biotin-SS31 pulldown ANT1 blot data, we normalized bands to the Biotin-SS31 group in each experiment. This eliminated the variation due to differences in sample loading amounts and blotting procedures.

For the ANT1 and UCP2 Western Blot data, we normalized the blotting bands to the total protein content in that lane, as detected by reversible protein stain. This eliminated variation from differences in sample loading amounts.

14) What was the justification for carrying out experiments in Figure 2 at pH 5.3? This pH does not seem relevant for general mitochondrial function. Do you see the same results at a pH close to actual mitochondria pH?

We appreciate this question. The mitochondrial intermembrane pH is around 6.9. While we did see a similar phenotype for proton entry at pH 6.9 (we have added this data as Figure 2—figure supplement 2 in this revised version and have noted this in the text), differences between proton permeability of young vs old inner membranes and between SS-31 treated and untreated were larger and less variable at pH 5.3. We believe that the observations at pH 5.3 are more clearly indicative of physical chemical differences in inner membrane permeability, in spite of this being a non-physiologic pH.

15) The Discussion made some conclusions that were maybe a bit too much speculation. For example, the proton leak is identified as pathological, although this was not directly demonstrated in the study. Also, protein conformation was inferred as well.

We thank the reviewer for this comment and have revised the language used. We have restricted the use of the term pathological to exclusively refer to previous observations of diastolic dysfunction of the aged hearts, which is not a healthy condition.

The conformational states in the ANT pore conferred by BKA and CAT treatment are based on prior work by others: PMID: 30611538 and PMID: 14603310, which are generally well accepted. We have changed the wording to make this reference clearer.

16) It should be emphasized that the proton leak studied is in myocytes in the resting, non-working state oxidizing endogenous substrates. The lack of a resting defect in OCR in this state is therefore plausible compared to the extensive literature describing key defects in OXPHOS with aging.

Thank you for this constructive suggestion. We emphasized the resting and non-working myocyte states in both the Materials and methods and Results sections.

17) The myocyte yield of the isolation procedure should be discussed. Are only the "best" aged myocytes obtained?

Each experiment, only preparations having at least 80% rod shape cells (which means live cardiomyocytes) were used. Thus, while we cannot completely exclude the possibility that there was some selection present, this would have to have been a small effect that could not have appreciably affected the reported results. For the aged myocytes, we did observe that the cells were on average larger, compared with the young cells, as has been previously documented by imaging of heart tissue sections by us and others.

18) The pH dependence of 405 excitation is of some concern. Also, calibration curves should be shown for pH 5.3 and 6.9; which are critical to the work.

Thank you for this suggestion. We added the pH calibration results of both 488 and 405 excitations in Figure 2—figure supplement 1. Also, we replaced the previous pH 4.5 and 6.0 panels with the calibration results at pH 5.3 and 6.9 in Figure 2—figure supplement 1.

19) The proton leak shown is of interest since state 4 rates in isolated cardiac mitochondria have minimal change with age. This should be discussed and reconciled.

An accurate State IV is difficult to achieve since there may be contaminating ATPases in the crude mitochondrial preparation which convert newly formed ATP back to ADP, and thus prevent a true “ADP exhausted” state. Thus, most investigators use oxygen consumption in the presence of oligomycin as a surrogate of state IV. This is the parameter that we have previously shown to be elevated in aged cardiomyocytes and reverse by SS-31 (PMID: 32648542).

20) The similarity of BKA and CAT to block proton leak is somewhat puzzling since the response of MPTP to the inhibitors is of course different. Is there evidence of MPTP opening in the permeabilized cells used for the measurement of mitochondrial pH with the buffer used as the artificial "cytosol"?

The discrepancy of BKA and CAT effect on proton leak and mPTP opening is reasonable, as ANT is only a regulator of the mPTP, while the ATPase itself, especially the "c" ring is thought by many to be the pore of the mPTP. Thus, while we hypothesize that both BKA and CAT change the conformation of the ANT pore in such a way as to inhibit proton entry, only one of these conformational states regulates the mPTP to slow its opening. In fact, this difference is a strong argument against the mPTP pore itself being the source of excessive proton leak in old cardiomyocytes, as CAT, which has no delay on mPTP opening times (Figure 5B), would not be expected to alter proton permeability, which it does.

21) The differences in Figure 2A are modest; the "normalized" data in Figure 2B and Figure 3C are more reasonable. How were fluorescence data normalized?

The differences seen by viewers in the fluorescence image are dependent on the observer’s display settings (for example γ) and may vary among viewers. On our displays and printed copies, the variation in intensities appear consistent with the quantified data, but the reviewer’s comment is an important reason for not relying on the visual image. To quantify the fluorescence intensity we first subtract the signal intensities in the background that does not contain any cell (this is generally low). Then, for each image area that contains a cell (ascertained by threshold settings and standard image segmentation algorithms) the integrated fluorescence value at the 488nm detection channel is divided by the corresponding value at from the 405nm detection channel. We set the ratio of 488/405 at the beginning as one (see query 13 above) and all later values are normalized to this starting value.

22) The dynamic range and reproducibility of the cp-YFP assay should be described in greater detail. It seems puzzling that some groups apparently required n=19 for valid results.

Thank you for the suggestion. We put the cp-YFP assay spectral characteristics at pH 5.3 in the Figure 2—figure supplement 1. The reproducibility of the data comparing old vs. young cells and old SS-31 treated cells at pH 5.3 for example (Figure 2C), yields a mean of 0.59, a SD of ± 0.10 and a range of 0.46 to 0.77 for the old compared to the starting pH 7.5 value. This range is non-overlapping with the mean of either young or old-SS-31 treated cells at pH 5.3. This is added to the Figure 2C legend to give a better understanding of the range and reproducibility of the assay.

The number n=19 arises because we used the results of multiple experiments to validate and confirm the data. It does not require a sample size of 19 to reach a significant difference in the statistical test.

23) Is there a quality control assessment of mitochondrial inner membrane integrity in the permeabilized cells at low pH? This is critical to the conclusion of the study.

As the mt-cpYFP indicator (269 aa, ~28kd) is still held within the mitochondrial matrix, this indicates that at the very least, the inner membrane does not break down enough to let this size molecule escape. Furthermore, and more to the point, while the reviewer is correct in thinking that a generalized breakdown of inner membrane integrity would prevent an informative answer as to the source of proton permeability, the fact that SS-31, BKA and CAT prevent excess proton permeability is a very strong indication that there is not a general loss of inner membrane integrity.